# Fifty Years of Tidewater Glacier Surface Elevation and Retreat Dynamics along the South-East Coast of Spitsbergen (Svalbard Archipelago)

**Jan Kavan [1,2,\*]**, **Guy D. Tallentire [3]**, **Mihail Demidionov [1]**, **Justyna Dudek [4]** and **Mateusz C. Strzelecki [2]**

1    Polar-Geo Lab, Faculty of Science, Masaryk University, 61137 Brno, Czech Republic;
     demidionovmihail@gmail.com
2    Institute of Geography and Regional Development, University of Wroclaw, 50-137 Wroclaw, Poland;
     mateusz.strzelecki@uwr.edu.pl
3    Geography and Environment, Loughborough University, Loughborough LE11 3TU, UK;
     g.d.tallentire@lboro.ac.uk
4    Institute of Geography and Spatial Organization, Polish Academy of Sciences, 00-818 Warsaw, Poland;
     j.dudek@geopan.torun.pl
\*    Correspondence: jan.kavan.cb@gmail.com

**Abstract:** Tidewater glaciers on the east coast of Svalbard were examined for surface elevation changes and retreat rate. An archival digital elevation model (DEM) from 1970 (generated from aerial images by the Norwegian Polar Institute) in combination with recent ArcticDEM were used to compare the surface elevation changes of eleven glaciers. This approach was complemented by a retreat rate estimation based on the analysis of Landsat and Sentinel-2 images. In total, four of the 11 tidewater glaciers became land-based due to the retreat of their termini. The remaining tidewater glaciers retreated at an average annual retreat rate of 48 m year$^{-1}$, and with range between 10–150 m year$^{-1}$. All the glaciers studied experienced thinning in their frontal zones with maximum surface elevation loss exceeding 100 m in the ablation areas of three glaciers. In contrast to the massive retreat and thinning of the frontal zones, a minor increase in ice thickness was recorded in some accumulation areas of the glaciers, exceeding 10 m on three glaciers. The change in glacier geometry suggests an important shift in glacier dynamics over the last 50 years, which very likely reflects the overall trend of increasing air temperatures. Such changes in glacier geometry are common at surging glaciers in their quiescent phase. Surging was detected on two glaciers studied, and was documented by the glacier front readvance and massive surface thinning in high elevated areas.

**Keywords:** tidewater glaciers; surface elevation changes; glacier geometry; Svalbard

## 1. Introduction

Glaciers cover almost 60% of Svalbard [1], constituting the predominant part of the local ecosystem. The interior of the archipelago is covered by vast ice caps often flowing directly into the ocean. This is also the case of the eastern coast of Svalbard, where a large part of the coastline is formed by fronts of marine terminating glaciers. Only small areas are formed by steep cliffs where glaciers are not present. Glaciers are essential stores of water in the Arctic, releasing the stored water during a relatively short melting period, usually between three to four months on the south-east coast of Svalbard [2]. Tidewater glaciers present on Svalbard's east coast are often considered as biological hotspots providing the adjacent marine environment with essential nutrients and other mineral material. Many animal species rely heavily on this element of the ecosystem, typically marine mammals or birds in Svalbard [3]. The inflow of nutrients also supports primary production near glacier calving fronts [4,5].

Blaszczyk et al. [6] identified 163 tidewater glaciers present on the Svalbard archipelago with a total length of 860 km calving ice-cliffs, while, Nuth et al. [1] identified 740 km of

tidewater glacier terminus width, only a few years later. Blaszczyk et al. [6] also stressed the importance of calving, which contributes as much as 21% to the overall mass loss from Svalbard glaciers. Hagen et al. [7] estimated the calving of tidewater glaciers to contribute slightly less (approximately 15%). Martín-Moreno et al. [8] reported a significant loss of glacier area, namely in the south and south-east of Svalbard, where a reduction of approximately 16% was estimated since the end of the Little Ice Age.

There is widespread agreement that Svalbard's glaciers have been losing mass since the Little Ice Age, with over 5000 km$^2$ of ice lost since the glaciers and ice caps of the archipelago were at their most recent maxima [8]. In this period, the greatest retreat rates were observed for tidewater glaciers which occurred in response to oceanic warming [9]. The last 50 years have seen an increase in the rate of mass loss of Svalbard's glaciers, showing an even greater negative mass balance since 2000, a result of Svalbard being one of the fastest warming regions on Earth [10]. Much of this understanding comes from in-situ mass balance measurements carried out over four decades on some of the region's glaciers, and is supported by remote sensing observations which also indicate increased glacier frontal retreat and mass loss [1,11]. Despite the recent advancement in knowledge about the response of Svalbard's glaciers to climate change, derived from these datasets, very little is known about the evolution of glaciers in Svalbard in the early 20th century [12]. Most of the mass balance observations and measurements on Svalbard have been carried out on small land-based glaciers in the west and central regions of the archipelago [7]. The areas most difficult to access, including the eastern and the southern coasts of Spitsbergen, are among the least studied parts of the Svalbard archipelago [13–16]. Direct mass balance measurements cover less than 0.5% of the glaciated area of Svalbard. The reference glaciers are on the west of the archipelago, and the climatic differences between the regions of study and a lack of similar in situ measurements on the eastern coast make it difficult to predict glacier evolution in this area during the same period [17].

Precise historic geographic data are rather sparse in the high Arctic and most of the information on glacier dynamics is usually limited to the last 20 or 30 yearswhen remote sensing data has become more readily available [18–22], or are limited to a single glacier where past direct observational data are available [23–25]. Members of the glaciological community are trying to bridge the gap by using different modelling approaches [12,26,27].

The aim of this paper is to describe the retreat rate and surface elevation changes of the eleven tidewater glaciers, as well as the overall development of glaciers on the east coast of Svalbard. To achieve this, we use historic satellite and aerial images, and a 1970 digital elevation model (DEM), which represents a unique opportunity to study long-term glacier dynamics.

## 2. Material and Methods

### 2.1. Study Site

The study site is the south-eastern region of Spitsbergen, the largest island of the Svalbard archipelago, and covers 77.5 km of coastline, with 11 glaciers terminating into the Barents Sea in 1970 (Figure 1). The total glacier area as defined in 2019 is 283 km$^2$. Note that the 1970 DEM does not cover the complete area of all the glaciers studied. The basic characteristics of all glaciers studied are summarized in Table 1.

The archipelago of Svalbard intersects three large water bodies: the Greenland Sea to the west, the Barents Sea to the east, and the Arctic Ocean to the north. The temperature of the surrounding water masses influences the climate of the archipelago, which is milder than that of other areas at the same latitude and, at the same time, more sensitive to changes relating to atmospheric fronts [28]. The eastern part of Svalbard experiences a cold climate characterized by relatively low temperatures as a result of the cooler waters of the East Spitsbergen Current flowing along its coastline, which often brings sea ice from the interior of the Arctic. During the Little Ice Age and in the 19th century, the eastern shores of Spitsbergen were often blocked by pack ice even during summer [14].

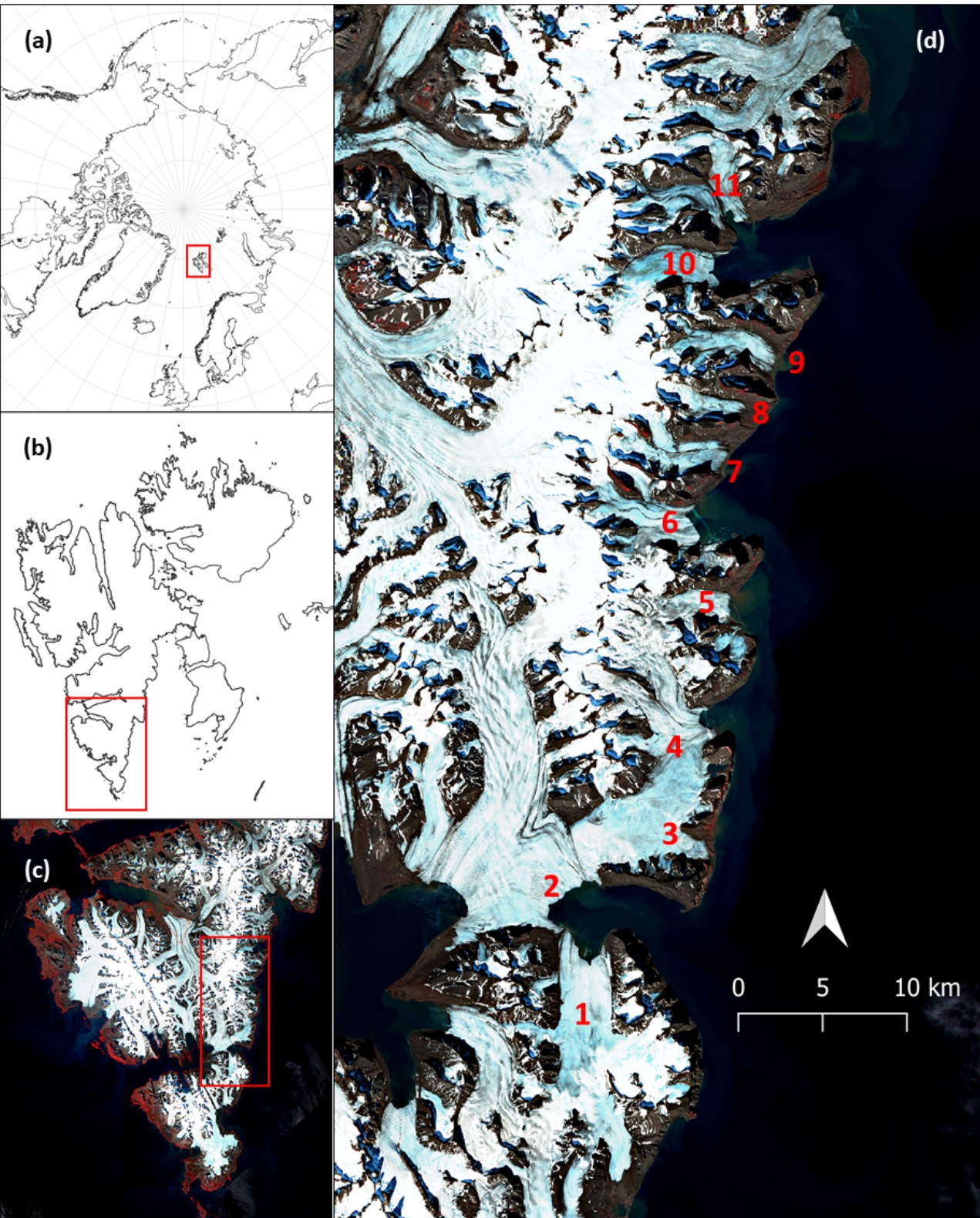

**Figure 1.** Study site; (**a**) the Arctic with location of Svalbard; (**b**) Svalbard with close up of the southern Spitsbergen island; (**c**) Sentinel-2 satellite image of southern Spitsbergen in false color; (**d**) the glaciers used in this study: 1—Sykorabreen, 2—Hambergbreen, 3—Staupbreen, 4—Markhambreen, 5—Crollbreen, 6—Davisbreen, 7—Skimmebreen, 8—Bellingbreen, 9—Anna Margrethebreen, 10—Emil'janovbreen, 11—Spaelbreen.

**Table 1.** Glacier characteristics (area and length derived from 2017 Sentinel images; elevation data based on 1970 DEM).

| Glacier | Id (Figure 1) | Area (km$^2$) | Length (km) | Mean Elevation (m) | Maximum Elevation (m) | Present Terminus |
|---|---|---|---|---|---|---|
| Sykorabreen | 1 | 54.6 | 16.7 | 248 | 550 | Marine |
| Hambergbreen | 2 | 16.6 | 13.2 | 189 | 360 | Marine |
| Staupbreen | 3 | 20.2 | 7.7 | 257 | 610 | Land |
| Markhambreen | 4 | 44 | 9.2 | 278 | 640 | Marine |
| Crollbreen | 5 | 20.4 | 8.8 | 279 | 630 | Marine |
| Davisbreen | 6 | 31.2 | 12.1 | 252 | 570 | Marine |
| Skimmebreen | 7 | 14.6 | 8.1 | 261 | 580 | Land |
| Bellingbreen | 8 | 6.7 | 6.6 | 266 | 570 | Land |
| Anna Margrethbreen | 9 | 14 | 8.4 | 208 | 550 | Land |
| Emil'janovbreen | 10 | 33.7 | 16.4 | 352 | 710 | Marine |
| Spaelbreen | 11 | 26.9 | 9.9 | 291 | 670 | Marine |

This stands in contrast with the western coast, where the West Spitsbergen Current brings warmer waters (due to its connection with the Gulf Stream) flowing from the south, restricting the duration of sea ice cover to a shorter temporal period [29,30]. These climatic differences are also well reflected in the presence of vegetation cover on the western coast compared to the eastern coast which has minimal vegetation (see Figure 1c) [31]. Such climatic characteristics result in the presence of vast ice caps and numerous glaciers which are marine terminating, making tidewater glaciers the prevalent glacier type on the east coast of Svalbard. Nordli et al. [32,33] reported the long-term air temperature series showing a distinct increasing trend since the 1970s, despite the timeseries being a composite for the whole of Svalbard. The eastern coast of Svalbard is not permanently monitored [28,29], but it is suggested that the air temperature has similar temporal patterns as the composite series shown by e.g., Sinnhuber [34] for the north-east region of Svalbard. The morphology of the terrain is heterogeneous with steep cliffs in the coastal areas reaching up to 500 m a.s.l., which are carved by large glaciers flowing down from the ice caps and plateaus towards the ocean. The highest peaks in the accumulation zone reach up to 800 m a.s.l.

The first modern comprehensive inventory of Svalbard glaciers was compiled by Hagen et al. [28], which constitutes the basis to describe glaciers included in this study. Blaszczyk et al. [6] specifically described Svalbard's tidewater glaciers with emphasis on flow velocity, calving fluxes, and mass loss. Many tidewater glaciers in Svalbard have experienced surge events during the last 150 years [13,35–40]. Such activity is described for most of the glaciers in this study [6,13], even though there is only one reported surge event on Markhambreen from around 2010. Noormets et al. [41] reported two major surge events prior to the study period at Hambergbreen. The eastern coast of Svalbard is inaccessible and thus lacks any long-term direct mass balance monitoring, with most of our knowledge based on remote sensing data often with restricted spatial and temporal resolution [10].

### 2.2. Detection of Surface Elevation Changes

Two digital elevation models (DEMs) were used to demonstrate the evolution of the glaciers' geometry. The first DEM was produced by Norwegian Polar Institute (NPI) and is derived from aerial images taken in the summer of 1970. The spatial resolution of the model is 20 m. The second DEM, produced by the Polar Geospatial Data Center at University of Minnesota (ArcticDEM), has a 2 m resolution [42]. The final Mosaic product (a compilation of several individual tiles) was used as the study site covered a relatively large area and no individual overflight (or series of overflights from one single year) was detected as covering the study site completely. This has obviously limited the precise quantification of the surface elevation changes, as no precise date could be attributed to the ArcticDEM. Through comparison with Sentinel-2 derived front positions of the glaciers, the ArcticDEM is most likely averaged to the state in the year 2017.

The ArcticDEM is projected to the National Snow and Ice Data Center (NSIDC) Sea Ice Polar Stereographic North and referenced to the ellipsoidal WGS84 horizontal datum

(EPSG:3413). The 1970 NPI DEM is projected using the European Terrestrial Reference System 1989 using the GRS 1980 ellipsoid (local reference system UTM-zone 33). As a result, the ArcticDEM shows a systematic shift in altitude of 28.5 m compared to the local reference system. The discrepancy between the two projection systems was solved by correction of the ArcticDEM. The shift of 28.5 m was estimated with use of 20 ground control points spread evenly throughout the study area. Comparison of the elevation values was made on flat surfaces in the study area in order to avoid errors resulting from the different resolutions of the DEMs. The standard deviation of the elevation shift was 1.03 m. The differences are likely to be more obvious in areas with high slope or heterogeneous terrain, which has to be considered when interpreting the results. Therefore, the quantification of the surface elevation changes serves mostly for identifying the spatial patterns of the changes and not for precise calculations.

The surface elevation changes were obtained as a simple overlap of the two DEMs, excluding the non-glacierized areas. The resulting surface elevation changes were used to define the zero-surface elevation change altitude as a parameter describing mass loss and gain. Glacier boundaries were defined using Sentinel-2 images from 2017 (frontal and lateral zones of the glacier) and ArcticDEM derived boundaries in the accumulation zone (based on flow accumulation function). The longitudinal profiles for each of the glaciers were derived in the centrelines and consisted of the profile derived from the 1970 DEM and ArcticDEM.

### 2.3. Glacier Front Positions

A series of Sentinel-2 and Landsat 5, 7, and 8 images were used to detect the positions of glacier fronts in 2000, 2010, and 2019. Positions of glacier fronts in 1990 were derived directly from the digital map database of the Norwegian Polar Institute. These were calculated based on a 1990 aerial mapping survey. Overall, two glaciers not covered by the photogrammetric overflight in 1990—Sykorabreen and Hambergbreen—and were instead mapped using Landsat imagery acquired in the summer of the same year. Glacier front positions in 1970 were derived directly from the 1970 DEM as described above. The satellite images were downloaded via the Sentinel Hub EO browser. The images were already georeferenced, and the TIFF files were directly processed in the QGIS environment. The positions of the glacier fronts were delimited manually for each of the years studied. Images obtained at the end of the summer seasons were used for the detection of front positions to keep consistency in the detection process and to avoid irregularities caused by seasonal variations in glacier front positions (winter advance/summer retreat).

The average retreat rate was calculated as a ratio of the area delimited between the glacier front positions studied and the length of the glacier centerline between the two front positions.

For more information on data sources refer to Supplementary Table S1.

## 3. Results

### 3.1. Glacier Retreat

All of the glaciers studied have experienced retreat since 1970 (Table 2). The average retreat rate was 48 m year$^{-1}$, with the maximum retreat rate found at Hambergbreen (up to 150 m year$^{-1}$). In total, four of the 11 glaciers studied have changed from marine terminating to land-based glaciers, which has also affected (i.e., slowed) their retreat rate (Figure 2). Staupbreen, Bellingbreen, and Anna Margrethbreen became land-based between 1970 and 1990, whereas Skimmebreen made this transition between 1990 and 2000. The high retreat rate of the Hambergbreen and Sykorabreen glaciers resulted in the separation of the original glacier into two almost independent glaciers and consequently also increased the width of the calving front of the two glaciers. Breakup of the original vast tidewater glacier into separate glacier termini is a relatively common feature and can also be observed in the case of Emil'janovbreen and Spaelbreen, where the two glaciers became separated

around 2005. Similarly, Davisbreen and Skimmebreen were terminating directly into the sea in one glacier terminus until around 1990.

**Table 2.** Surface elevation changes and retreat rates of the glaciers studied; average surface elevation change uncertainty is ±1.1 m, the annual change uncertainty corresponds to ±0.023 m/year; ice front retreat rate uncertainty is ±0.8 m.

| Glacier | Id (Figure 1) | Average Surface Elevation Change (m) | Annual Change (m/year) | Ice Front Retreat Rate (m/year) |
|---|---|---|---|---|
| Sykorabreen * | 1 | −12.7 | −0.27 | 149.0 |
| Hambergbreen * | 2 | −39.2 | −0.83 | 149.0 |
| Staupbreen | 3 | −29.9 | −0.64 | 16.3 |
| Markhambreen | 4 | −30.5 | −0.65 | 24.5 |
| Crollbreen | 5 | −10.7 | −0.23 | 32.7 |
| Davisbreen | 6 | −11 | −0.23 | 89.8 |
| Skimmebreen | 7 | −19.5 | −0.41 | 21.4 |
| Bellingbreen | 8 | −13.8 | −0.29 | 20.4 |
| Anna Margrethbreen | 9 | −23.8 | −0.51 | 10.2 |
| Emil'janovbreen * | 10 | −45.2 | −0.96 | 71.4 |
| Spaelbreen * | 11 | −24.1 | −0.51 | 71.4 |

* one value for both glaciers as they share the common outlet.

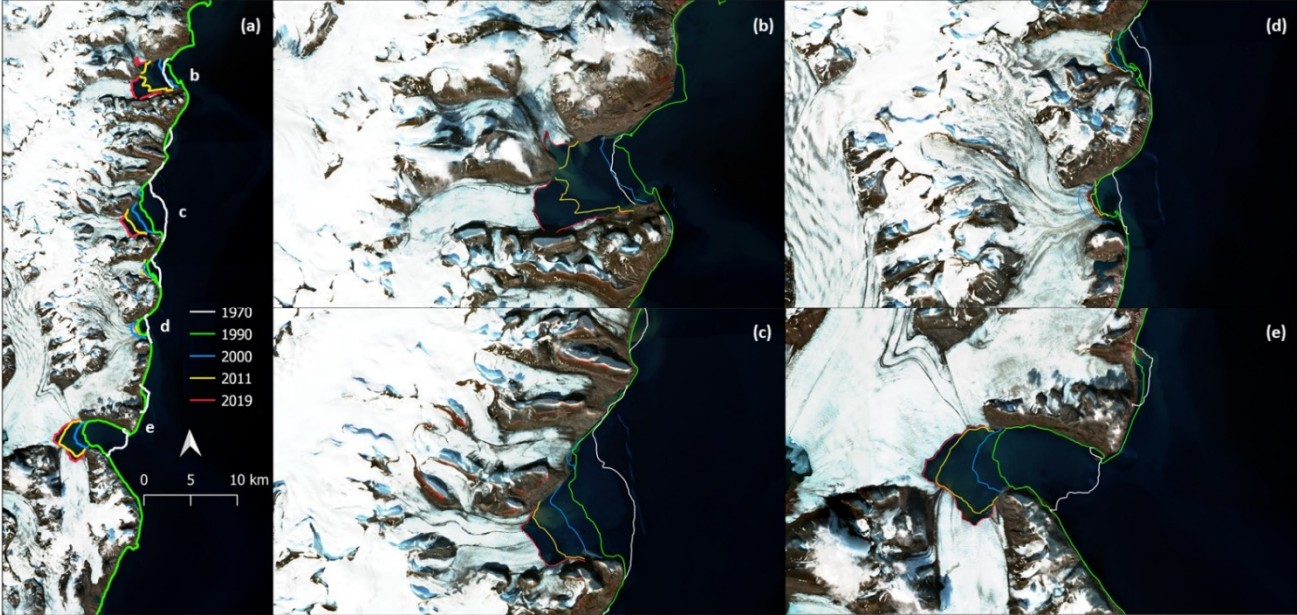

**Figure 2.** Glacier front positions in 1970, 1990, 2000, 2010, and 2019 of the tidewater glaciers that remained marine terminating in 2019; the whole study site (**a**) with close up views of four major areas where the retreat was quantified. (**b**) Emil'janovbreen and Spaelbreen, (**c**) Davisbreen and Skimmebreen, (**d**) Markhambreen and Crollbreen, (**e**) Hambergbreen and Sykorabreen.

The irregularities found in the retreat rate can be attributed to natural variability or occasional surge events. A minor readvancement was recorded in the case of a common outlet of Emil'janovbreen and Spaelbreen between 1970 and 1990; although we do not have any written record of a surge event, the high surface elevation loss would suggest that this took place. In the case of Markhambreen, a surge event was recorded around 2010, and is quite clear from the glacier front positions illustrated in detail in Figure 3.

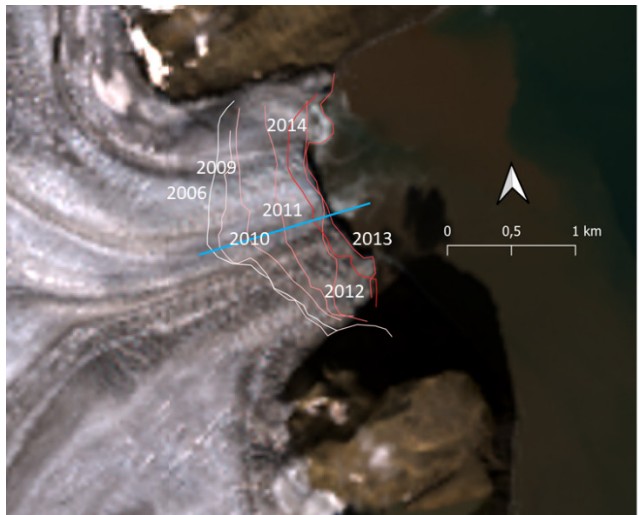

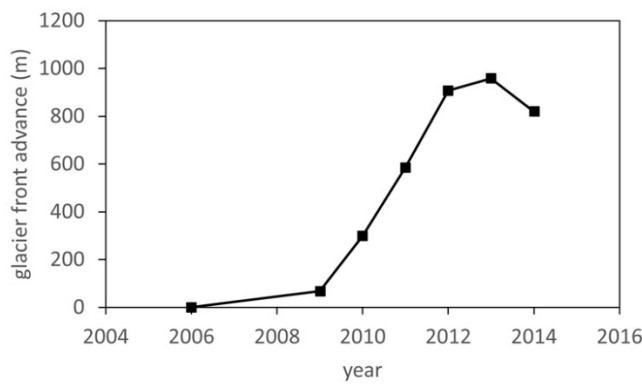

**Figure 3.** Glacier front positions during the surge event on Markhambreen with the advance rate in graphics, Landsat image from 2013 as a background image.

### 3.2. Surface Elevation Changes

All of the glaciers studied experienced similar development in terms of changes in their surface elevation between 1970 and the present. The common feature that can be observed is the surface lowering in their ablation areas, which is apparent up to an altitude of approximately 200–300 m a.s.l. The higher reaches of the glaciers experienced no clear change or gain in their mass. However, this trend is disturbed by occasional surge events as reported, for example, from Markhambreen since 2008. The readvancing of the glacier front is visible in Figure 3, and the disturbed post surge geometry is also well documented in Figure 4. Similarly, Emil'janovbreen experienced a surge event between 1970 and 1990, as documented by the readvancing glacier front (Figure 2) and massive surface lowering reaching up to 120 m and propagating towards the glacier terminus (Figure 4). Markhambreen experienced severe loss in mass in the whole northern branch of the glacier i.e., surface lowering in the accumulation zone. The two minor branches of the glacier system did not undergo surging and saw slight mass gain during this time. The surface elevation changes are summarized in Table 2.

### 3.3. Zero Surface Elevation Change Altitude

The altitudes which distinguish mass loss and gain vary among the glaciers studied from 240 to 530 m a.s.l. (Table 3). The estimated zero-surface elevation change altitude (ZSECA) could be considered as the long-term average for the whole study period. There are clear differences among the glaciers, but the general trend is dominant. The lower parts of the glaciers have lost substantial mass, whereas the upper parts remained relatively stable or even gained some mass-up to 20 m in some cases (Figure 5). The two exceptions were the glaciers that experienced surges during the study period, i.e., Markhambreen and Emil'janovbreen, where the whole glacier surface experienced thinning. Some of the glaciers were not fully covered by the 1970 DEM. It was therefore not possible to estimate the ZSECA for Hambergbreen. We can only suggest that all the glacier is currently below the ZSECA as all the surface covered by 1970 DEM is thinning, and the small part not covered by the 1970 DEM lies below 200 m a.s.l.

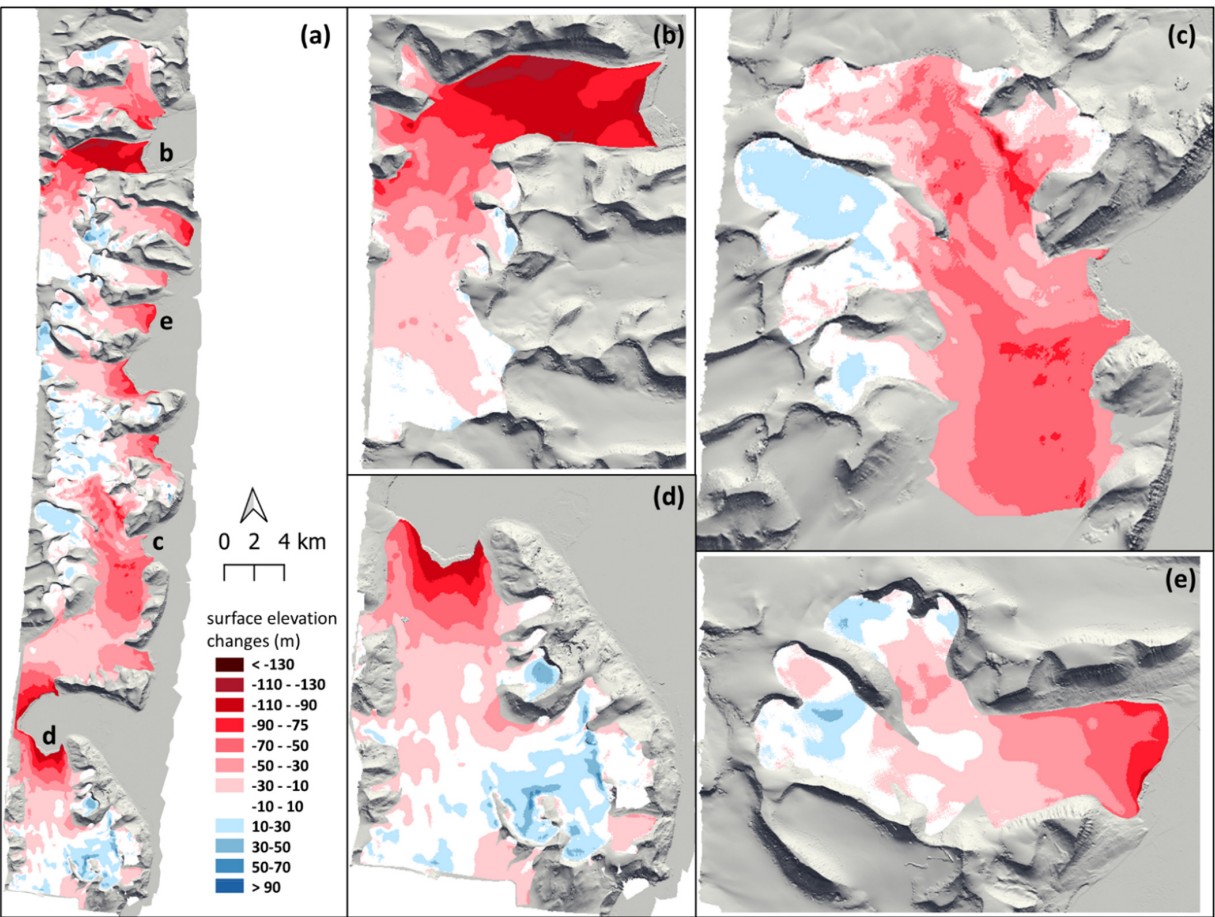

**Figure 4.** Surface elevation changes between 1970 and recent ArcticDEM expressed as elevation change (**a**); the detailed views represent the glaciers frequently mentioned in the text i.e., the two glaciers that underwent surge (**b**) Emil'janovbreen, (**c**) Markhambreen; (**d**) Sykorabreen as a part of the Hambergbreen-Sykorabreen glacier system with the highest retreat rate and (**e**) Skimmebreen as an example of a glacier that turned into land-based system; background is the ArcticDEM, glacier outlines delimited from 2017 Sentinel-2 images.

**Table 3.** Zero surface elevation change altitude (ZSECA) and glacier above ZSECA ratio; the uncertainty of area above ZSECA values arisen from the elevation uncertainty are reported in brackets.

| Glacier | Id (Figure 1) | ZSECA (m a.s.l.) | Glacier above ZSECA (%) | Comment |
|---|---|---|---|---|
| Sykorabreen | 1 | 230 | 46.8 (42.2–49.7) | |
| Hambergbreen | 2 | No | 0 | The whole glacier is losing mass |
| Staupbreen | 3 | 480 | 1.2 (0.8–1.3) | |
| Markhambreen | 4 | 350 | 8.5 (6.6–15.3) | Southern part only, northern losing mass |
| Crollbreen | 5 | 240 | 45.4 (39.7–50.6) | |
| Davisbreen | 6 | 250 | 45.7 (42.1–47.6) | |
| Skimmebreen | 7 | 300–400 | 18.6 (15.9–21.2) | Different altitude on different glacier branches |
| Bellingbreen | 8 | 370 | 17.7 (16.2–21.9) | |
| Anna Margrethbreen | 9 | 250 | 23.9 (21.8–25.9) | |
| Emil'janovbreen | 10 | 530 | 9 (7.3–11.2) | Southern part only |
| Spaelbreen | 11 | No | 0.2 (0–0.6) | Small discontinuous areas of mass gain |

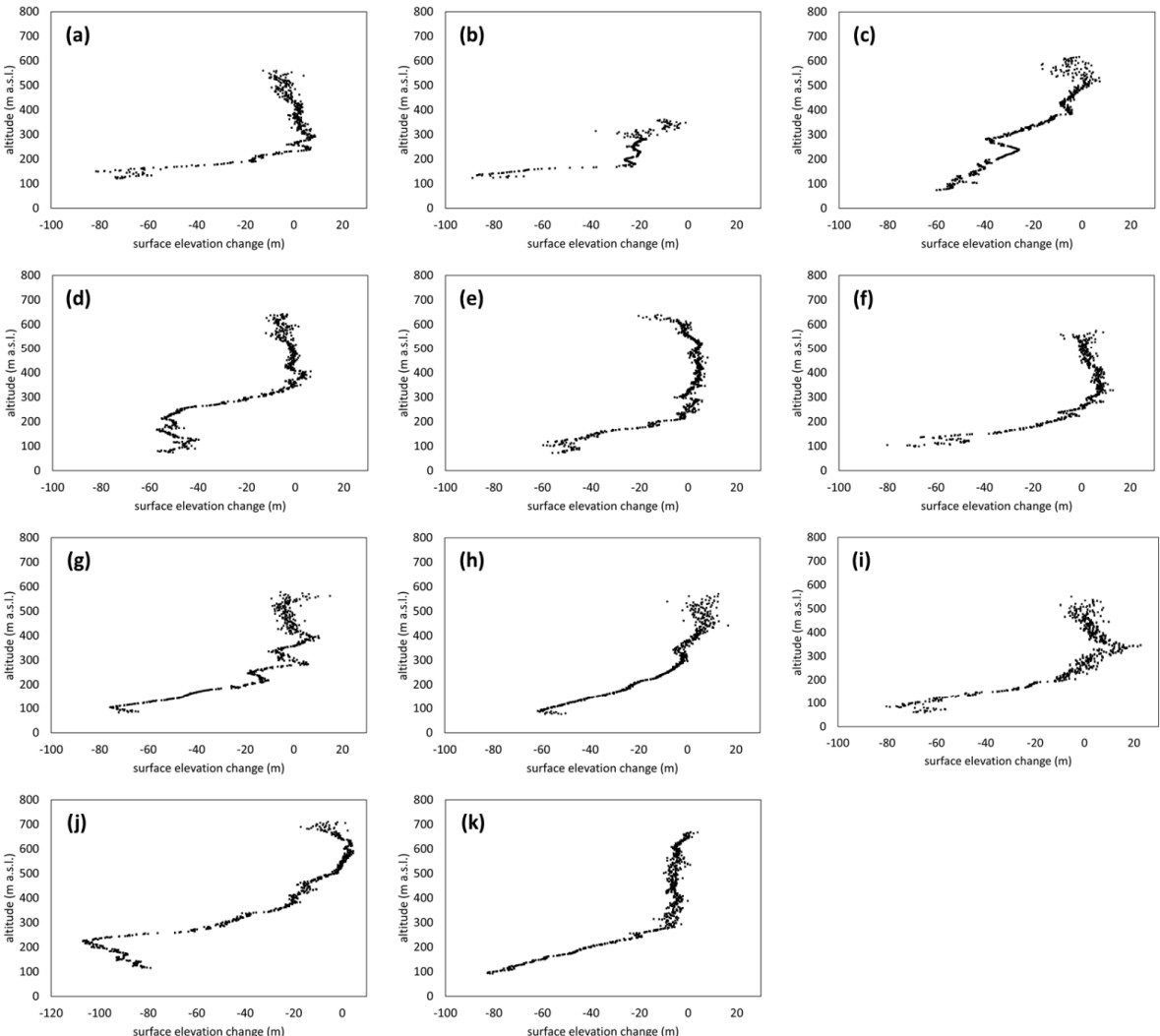

**Figure 5.** Surface elevation changes in relation to glacier altitude (1970 DEM); (**a**) Sykorabreen, (**b**) Hambergbreen, (**c**) Staupbreen, (**d**) Markhambreen, (**e**) Crollbreen, (**f**) Davisbreen, (**g**) Skimebreen, (**h**) Bellingbreen, (**i**) Anna Margrethebreen, (**j**) Emil'janovbreen, (**k**) Spaelbreen.

### 3.4. Glacier Geometry

The changes in glacier thickness combined with massive retreat of the glacier fronts resulted in significant change in glacier geometry. A number of the glaciers retreated up to 7 km (e.g., Hambergbreen) and significantly reduced in thickness at their calving front. On the contrary, most of the glaciers experienced mass gain in the highest parts of their accumulation areas, which has led to an increase in slope. In general, the present glaciers have steeper surfaces, are shorter in length, are thinner in their ablation zones, and have thicker accumulation areas. The exceptions are Markhambreen and Emil'janovbreen. The northern branch of Markhambreen experienced a surge around 2010, and the entire northern part of the glacier has lost large portion of its mass, even in the high elevation areas. The same is true for Emil'janovbreen, which has lost a large amount of mass across its whole area and therefore subsequently experienced dramatic thinning. The comparison of centerline profiles between the 1970 DEM and ArcticDEM is illustrated in Figure 6 with examples from several of the studied glaciers. It is clearly visible that the lower parts of the glaciers are thinning, whereas the upper parts are gaining mass or do not change considerably. The exceptions of Emil'janovbreen and Markhambreen are also well illustrated. The central branch of Markhambreen experienced mass gain in its upper reaches, whereas the northern branch has lost much of its mass due to surging.

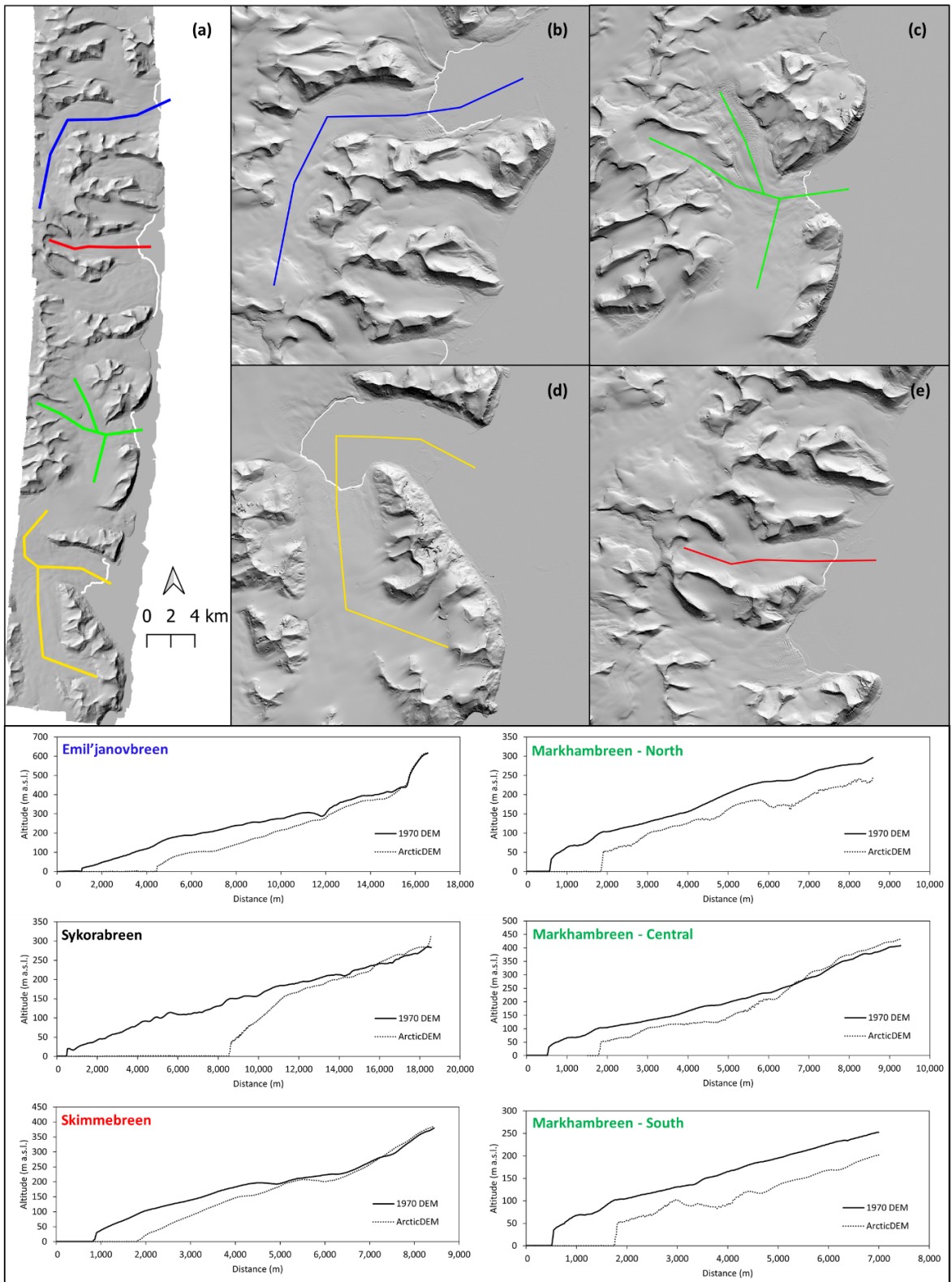

**Figure 6.** Longitudinal profiles (colored lines) of glaciers studied from DEM in 1970 and ArcticDEM; the overview of the whole study area (**a**); Emil'janovbreen (**b**); Markhambreen (**c**); Sykorabreen (**d**); Skimmebreen (**e**); selected glaciers shown in this figure correspond to the glaciers from Figure 4; The overview figure with 1970 DEM as background, the detailed views of profile lines with ArcticDEM as a background; The captions of the longitudinal profiles are in the same color as on the map to ease the orientation.

## 4. Discussion

The massive retreat rate of tidewater glaciers goes hand in hand with findings of glacier behavior from other parts of Svalbard ([10] for recent glacier behavior or Martín-Moreno et al. [8] for post Little Ice Age development), and specifically for other tidewater glaciers in the archipelago [6]. There are very few studies which have presented results of glacier recession that this could be related to. Historical analysis of glacier changes in this area showed that frontal retreat of Hambergbreen (ca. 16 km) was accompanied by its surface thinning of 60–100 m during the period of 1901–2000 [43]. The most rapid retreat rate (150 m year$^{-1}$) identified in the case of Hambergbreen is similar to that of Hornbreen, flowing towards the other side of the glaciated peninsula [44,45]. In fact, despite a recent slowdown of retreat in both Hornbreen and Hambergbreen, it is expected that the waterway between the east and west coast will open around 2055–2065 [43,46]. The retreat rate is lower in the case of glaciers that have already retreated and transformed into a land-based systems. In such cases, the lowering of the glacier surface is the dominant process as the frontal area is relatively thick. This behavior is similar to that of land-based glaciers right after their maximum extent at the end of the Little Ice Age [47].

All of the glaciers have lost large parts of their mass despite a slight surface elevation gain in the accumulation zones. This corresponds well with the increasing air temperature trend [32,33] that began in the 1970s across the whole of the Svalbard archipelago. The mass gain in the glaciers' upper reaches could be attributed to increased precipitation. Førland et al. [48] identified significant increase in precipitation at all weather stations used for assessing the precipitation trends between 1975–2011. This trend was even more pronounced in more recent years. The increase in precipitation does not compensate the increase in melt rates, and the annual surplus from the accumulation area will take some time to get the advected ice down to the ablation area. This is the case of most of the glaciers studied. This stems well with observations of Blaszczyk et al. [19] and Schuler et al. [10], who reported high velocities only on Hambergbreen and the lower part of Sykorabreen—the glacier with largest surface elevation losses. The mass loss across all parts of Hambergbreen, despite its snow line of 398 m a.s.l. as identified by Laska et al. [49], might also be explained by high flow velocities moving the ice mass downglacier from the accumulation area. The redistribution of snow due to the combination of terrain morphology and prevailing wind may also be an important factor [21,50]. This may be the cause for mass loss in the uppermost parts of Staupbreen and Crollbreen, for example (Figure 4). The comparison of glacier front positions in 1936 (as seen from the oblique aerial images) and 1970 (the DEM used in this study) suggested that the retreat rate was considerably lower before 1970, which may be explained by relatively stable air temperatures, including a short period of cooling during the 1960s [33], and the intensification of warm water inflow in the last decades which has resulted in increased ocean temperatures and a decline in sea ice extent [51]. The geometry of the glaciers probably led firstly to surface lowering, followed by retreat of the frontal area later, as reported for the first half of the 20th century in central Svalbard [46]. A similar change in glacier geometry was also reported by Marlin et al. [52] in the case of Austre Lovénbreen on the west coast of Svalbard. Moreover, the generally low altitude of the glaciers studied could have contributed to their massive ablation and could be a result of high sensitivity to air temperature variations [53]. Blaszczyk et al. [19] reported surface lowering ($-1.6$ m year$^{-1}$ between 2012–2017) at Hornbreen, the neighbouring glacier to Hambergbreen, in an area of the glacier where similar surface lowering was detected by this study. Strozzi et al. [54] detected the same spatial pattern of surface elevation changes in the case of Stonebreen tidewater glacier on the Edgeøya island (eastern Svalbard) when comparing a similar time period (1971–2011). Pälli et al. [43] also reported thinning in almost all of the studied glaciers since 1900. Surprisingly, Sykorabreen was gaining mass in the upper parts in the early 20th century.

Using the two DEMs which were acquired with different techniques (i.e., aerial photographs and satellite images) can bring some constraints to the precision of the data especially when comparing the two DEMS. However, regarding the long time span be-

tween the two DEMs and large differences in the elevation changes, only the general spatial patterns of change can be drawn. Similarly, the methods used in Pälli et al. [42] for estimating the changes on local glaciers have their limitations, resulting from the accuracy issues with historic mapping.

The described geometry changes in the glaciers (retreat and steepening of the accumulation zone) are classic signs usually detected before surge events [55]. Unlike land-based glaciers, where the surge usually begins in the accumulation zone, the surge of a tidewater glacier in Svalbard often starts at the terminus and propagates upwards [56]. This is very likely the case of the Markhambreen surge described here. The glacier has advanced approximately 900 m between 2009 and 2012, resulting in a mean annual advance of about 300 m. Such retreat and advance rates are in good correspondence with previously reported rates for example at Blomstrandbreen, in the northwest of Svalbard [57] when accounting for mass loss due to calving, which is estimated at around 20% [58]. This may have led to the flow velocity being in the correct window of reported values for most Svalbard glacier surges—around 1 m day$^{-1}$ [59]. Interestingly, the surge of Markhambreen coincides with the surge of Nathorstbreen (flowing to the north-east from the accumulation area neighboring the glaciers studied) despite the two glaciers not being directly linked. Sund et al. [58] reported that Nathorstbreen started surging after October 2008, and resulted in up to 50 m surface lowering in some areas. The mass loss in the accumulation area of Markhambreen was even more extreme, with certain areas of the glacier surface reducing by more than 70 m. Nuth et al. [60] used the example of the Nathorstbreen surge to demonstrate the impact of warming in the Arctic on the surging activity of local glaciers. Surprisingly, the thinning of the glacier tongue and subsequent basal freezing may enhance storage of subglacial meltwater in the upper parts of the glacier and thus lead to more surging. This may be the case of the glaciers studied, as all of them display the described parameters. Surging events were recorded also prior to the study period for example, at Hambergbreen [13], where recently Noormets et al. [41] detected two surges around 1900 and 1957. It is likely that the glacier front in 1970 (i.e., the basis for our comparison) represents the maximum extent after the surge starting in 1957. This may also explain the fastest recorded retreat rate (150 m year$^{-1}$) of the glaciers studied.

Tidewater glaciers do not only represent a dominant landscape feature on the eastern coast of Svalbard, but also constitute one of the most important ecological hotspots where the marine ecosystem is fed by nutrients and other compounds from the terrestrial environment [4,5,61]. The subglacial meltwater discharge also modifies the general circulation of water in front of the glacier terminus, usually leading to water upwelling and increased mixing of the fjord water [62,63]. This enhances the biological importance of such sites especially for marine mammals, or for foraging birds [3,64]. In this light, the shift of four glaciers from marine terminating to land-based can be seen as an important loss of biodiversity, as the fjord circulation at the glacier fronts will likely have changed, probably causing loss of certain species directly connected to the activity that occurs in this area (e.g., bearded seals). Similar development is also expected across other parts of Svalbard [65].

The observed glacier retreat also led to another significant change in the environment—opening of new bays and formation of pristine coastal zones. Such new coasts usually evolve from young unconsolidated glacial sediments left along the shores of fjords by retreating glaciers [45]. For instance, the opening of the Hambergbukta exposed over 10 km of fresh paraglacial coasts, where lateral moraines, crevasse-squeeze ridges, and eskers evolved into new narrow beaches, cuspate barriers, spits, and lagoons similar to those observed in the Brepollen on the opposite side of the Hornbreen-Hambergbreen glacier saddle [45]. Formation of the new embayment between the retreating Emil'janovbreen and Spaelbreen resulted in development of over 12 km of new shorelines filled with a number of well-developed accumulative coastal landforms, including one of the longest spit systems in this part of the archipelago, which formed along the northern banks of the new bay.

Overall, the coastal systems developing in front of retreating tidewater glaciers on the southeastern coast of Spitsbergen are one of the most geomorphologically dynamic paraglacial coastal environments in the entire Svalbard archipelago.

## 5. Conclusions

The tidewater glaciers in eastern Svalbard have retreated at a considerable rate (mean retreat rate 48 m year$^{-1}$) between 1970 and 2019, with a maximum retreat rate recorded of 150 m year$^{-1}$ at Hambergbreen. The retreat was so extreme that four of the 11 glaciers studied had completely lost their connection with the sea and became land-based glacier systems retreating from their respective fjords. The slowdown of the retreat at Crollbreen and Spaelbreen and appearance of new land in their lateral marginal areas suggests that these two glaciers may be the next to become land-based systems in the study area. It is highly probable that such behavior will be observed in many of the remaining tidewater glaciers in Svalbard in the near future. The shift from marine terminating to land-based glaciers may also have serious implications for biodiversity, as a number of species rely directly on processes at the glacier terminus i.e., calving, whilst many others benefit from fjord water circulation induced by meltwater discharge at a glacier's grounding line.

There was a common spatial pattern in the geometry of all of the studied glaciers with thinning in the ablation areas of the glaciers, and the gaining of mass in the upper accumulation areas. This together with the massive retreat has completely changed the geometry resulting in shorter and steeper glaciers. Such dynamics are often observed in glaciers in their pre-surge period. This may be the case of the glaciers studied, as it was also demonstrated at Markhambreen, which experienced an important surge around 2010. The surge event was also imprinted in the general surface lowering of all the northern part of the glacier and episodic readvance of its calving front. A similar behavior was observed at Emil'janovbreen, where surging occurred between 1970 and 1990. However, it is unlikely that all the glaciers will surge as the reduced glacier flow velocities may explain such geometry changes as well. The accumulation rate in the upper parts of the glaciers is far below the overall ablation rate. The glacier geometry changes and retreat rate reflected changes in climate well—higher air temperatures leading to surface lowering in the frontal zones and higher precipitation (together with reduced flow velocities) causing the mass gain in upper reaches.

The paper has shown the importance and great value of historic spatial data such as the 1970 DEM that has enabled us to assess the changes of glacier dynamics over almost half a century. The massive retreat of the glaciers has also exposed large number of new coasts that are currently being reshaped by coastal processes, and which require further work to understand their impact on southeast Spitsbergen and more broadly across the archipelago of Svalbard.

**Supplementary Materials:** The following supporting information can be downloaded at: https://www.mdpi.com/article/10.3390/rs14020354/s1, Table S1: Source data used in the analyses.

**Author Contributions:** Conceptualization, J.K.; methodology, J.K. and M.D.; software, J.K. and M.D.; validation, J.K.; formal analysis, J.K.; investigation, J.K.; resources, J.K. and M.C.S.; data curation, J.K.; writing—original draft preparation, J.K., G.D.T., J.D. and M.C.S.; writing—review and editing, J.K. and G.D.T.; visualization, J.K. and M.D.; supervision, M.C.S.; project administration, J.K.; funding acquisition, J.K. and M.C.S. All authors have read and agreed to the published version of the manuscript.

**Funding:** This research was funded by the Masaryk University project ARCTOS MU (MUNI/G/1540/2019) and MUNI/A/1570/2020. The research was also funded through the Norwegian Financial Mechanism 2014–2021: SVELTA—Svalbard Delta Systems Under Warming Climate (UMO-2020/37/K/ST10/02852) based at the University of Wroclaw.

**Data Availability Statement:** ArcticDEM data were provided by the Polar Geospatial Center under NSF-OPP awards 1043681, 1559691, and 1542736. NPI 1970 DEM was provided by the Norwegian Polar Institute.

**Acknowledgments:** This work was supported by the Masaryk University project ARCTOS MU (MUNI/G/1540/2019) and MUNI/A/1570/2020. The research leading to these results has received funding from the Norwegian Financial Mechanism 2014–2021: SVELTA—Svalbard Delta Systems Under Warming Climate (UMO-2020/37/K/ST10/02852) based at the University of Wroclaw. Arctic-DEM were provided by the Polar Geospatial Center under NSF-OPP awards 1043681, 1559691, and 1542736. We would like to thank to Veijo Pohjola and four anonymous reviewers for their valuable comments and suggestions.

**Conflicts of Interest:** The authors declare no conflict of interest.

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
