# Peer review of "Fifty Years of Tidewater Glacier Surface Elevation and Retreat Dynamics along the South-East Coast of Spitsbergen (Svalbard Archipelago)"

_remotesensing, doi:10.3390/rs14020354_

Round 1
Reviewer 1 Report
The authors study the evolution of glaciers on the east coast of Spitsberg (Norway) between 1970 and 2018. The study is based on both digital elevation models, first from the Norwegian Polar Institute and the second from Polar Geospatial Data Center at the University of Minnesota, and a series of Landsat and Sentinel images. The study is complete, addressing the evolution of glacier retreat, elevation changes, equilibrium line altitude and glacier geometry.
In an overview of the paper the content is well introduced, the objectives are clear and the study is very interesting, but there is a set of problems in the form (see comments) that disturbs the reading.
Regarding the scientific content, I'm puzzled about the elevation measurements (elevation changes, equilibrium line altitude). It's well known the maximum of errors in a digital elevation model is on the up axe, especially when the DEM is built from air born acquisition, moreover, when the image series are grabbed at different days. A drift appears on the up axe, in the areas where there is no ground control points (GCP), that's why GCP are used to geo-reference DEM. So, without GCP on at least one of the glaciers, it is difficult to validate the elevation measurements. Concerning the DEM registration it would be interesting to have the localization, relative altitude and shift of each of your GCP. Are they real GCP, measured in-situ, or are they registration points, picked on a DEM ? You explain the ArcticDEM uses a different projection system, but what system is used? Is the difference of elevation is natural according to the references systems? Is it a simple problem of geoide and elipsoide ?
Some comments
From lines 29 to 50 split in paragraphs is strange.
Homogeneize the Figure references
Remove the white spaces after/before "(" and ")"
Line 90, remove line break
Line 117 Add a white space after, "e.g."
Remove double white spaces
Line 135: I suppose Table 2017 is Table 1... Remove the table name on the screen shot of the table and give it a coherent name with the text references.
Figure 1: your subfigures have no name (a, b, c), it's not easy to make the link with the caption, and I suggest you use the same name for subfigures in all figures and captions e.g. (A, B, C...) or (a, b, c...)
line 184: it seems to white space missing before "(" and the exponent "-1" of the unit on 2 line is not readable
Figure 2 line 201: naming error "xbreen and xbreen" and 2 different writing "x breen"
Figure 4: the scale level +-1 is clearly not significant due to the elevation errors of both DEM and the error of DEM alignment.
Figure 5: don't switch the caption style
Keep the same naming style for tables, don't switch between 1, I, 2, II...
Table II How the Spael has no ELA and has a part above ELA?
line 259 figure 6 ->Figure 6
Author Response
dear reviewer,
first of all, thanks for your comments and suggestions which certainly helped to improve the quality of the manuscript. We provide detailed replies to all reviewers in the pdf file attached. We hope that the changes made and explanation of some issues will be satisfactory.
with regards,
Jan Kavan

Reviewer 2 Report
This paper shows an interesting multi-temporal study of tidewater glaciers in Svalbard, the subject is interesting and fits well to Remote Sensing scope. In my opinion the paper deserves for publication after some minor changes, discussions or explanations, bellow, I detail in some comments some aspects to improve:
-The submitted version is not clean, there are blanks, colours highlighting some parts of the text, etc. for example line 90-91.
-I do not recommend starting the title with a number, better Fifteen years…
-Tables shown are just images of tables, we recommend to adapt the table to Remote Sensing format, including font, size, etc.
-A table with the datasets used, their dates and their function (e.g. delineation of glacier front, input for volume changes estimation, etc.) is mandatory.
-I miss many methodological details, particularly those specifying DEMs errors-accuracies. Are these errors used in the volume estimation? In that case are you using a minimum level of detection or a spatially variable error based on any other parameter-technique? Additionally it would necessary to transfer these errors to estimations.
-I see some methodological problems with the Arcti DEM, please provide an average date, error estimations, etc.
-Your DEMs of difference approach could use stable non-eroded or sedimentation areas to see what changes are experienced there according to your approach and using this value as an error estimation for changes.
-Avoid graphs over images, e.g. Figure 3, provide just a white or black background for these graphs.
-Figure 4 provide scales.
-Figure 6 provide scales for some figures, provide legend for lines.
-I think the discussion about the processes and the evolution of the glaciers is fine, but I would recommend to add a paragraph being critical with the limited quality and possibilities of your dataset.
Author Response

(The authors gave the same response as above.)

Reviewer 3 Report
The authors map both elevation change and terminus position on 11 southeast Svalbard glaciers from 1970-2019. There is a consistent pattern of extensive thinning exceeding 50 m on all but one glacier (9). There is no significant thickening +10 m in the upper part of just three glacier s (6, 8 and 9). Only #4 did surge in this period. In terms of dynamics the value of this paper can be in distinguishing glaciers where a surge state maybe affecting dynamics and those where climate impact is dictating dynamics. The latter maybe the case even if the glacier can surge.
18: Be more specific than certain parts. Maybe rewrite “All the glaciers studied experienced thinning of more than 50 m in their frontal zones with maximum surface elevation loss exceeding 100 m in certain parts”
19: Minor is less than how much thinning? Minor thickening upglacier combined with major thinning downglacier is not a typical surge pattern. There should be a more substantial and widespread increase. Rewrite suggestion “In contrast to the massive retreat and thinning of the frontal zones, a minor increase in ice thickness was recorded in some highly elevated parts of the glaciers, exceeding 10 m on three glaciers.”
185: List the four glacier that ceased to be marine terminating and also what interval did that happen during?
188: The separation increased the width of the calving front of the two glaciers
227: Make clear how the ELA was derived and what it represents. It appears you used long term elevation data, which does not identify the area of the ablation zone. How does this compare to the observable snowline on the glacier? Contrast to Laska et al (2017) who note the ELA of Hornbreen at 398 m.
Table 2: The entire area of thinning in the accumulation zone is not necessarily ablation, this can be dynamic thinning too. For example, Hambergbreen does retain some snowcover each year.
Figure 6: Include the ice front on this image given the lack of color change at this transition.
288: Worth noting that the increased accumulation due to higher precipitation would also not be distributed downglacier as readily as the thinner glaciers have slowed down. Skykorabreen and Hambergbreen do not show any thickening even high on the glacier. This fits with velocity observations of these glaciers that show consistently high velocity near the front of both and all the way upglacier on Hambergrbreen (Blaszczyk et al 2019; Schuler et al 2020).
329: This is at odds with what is described in the previous section regarding increased precipitation (snowfall). Murray et al (2003) note that the thickening associated with other surge events exceeds that of what is observed here. Hence to surging as the mechanism these would have to be terminus region induced surges, where the quiescent period results in reduced flow upglacier, but in an environment of rising ELA this would limit actual surface thickening.
348: How many of the current marine terminating glaciers now are close to becoming land based? This could be indicated at line 370 instead.
363: Is this new coastline appealing to any birds or mammals?
379: It is unlikely that all are due to surging so make sure to note the other option of increased ablation low on the glacier and increased accumulation high on the glacier in an environment with reduced velocities. Schuler et al (2020) provide glacier velocities for some of the glaciers you examine, for Hambergbreen the velocity observations that are consistently high and do not support a recent surge state.
Laska, M.; Barzycka, B.; Luks, B. Melting Characteristics of Snow Cover on Tidewater Glaciers in Hornsund Fjord, Svalbard. Water 2017, 9, 804. https://doi.org/10.3390/w9100804
Author Response

(The authors gave the same response as above.)

Reviewer 4 Report
In general, the article is not very revealing. It points to the retreat of glaciers as a result of climate change - this is obvious and widely acknowledged.The authors indicate that the novelty is the use of two terrain models from the 1970s and the current one.
And here I have a major concern, which please explain.
Measurement techniques have changed over 50 years and are they indiscriminately comparable in this context? Going further, can detailed conclusions be drawn about changes in glacier morphometry on this basis?The use of DEMs to assess glacier change on Spitsbergen is not new either. Among other things, the work lacks reference to historical analysis of glacier change.
Glacier changes in southern Spitsbergen, Svalbard, 1901- 2000, 2003
All figures - start a sentence with a capital letter, not a hyphen.
Figure 2- 1970 status, please mark with a colour other than white- it is not readable nowadays.
Author Response

(The authors gave the same response as above.)

Round 2
Reviewer 1 Report
All my comments, remarks and corrections have been integrated which improves style and content, and makes the paper more readable. Thus, in my point of view, the paper is acceptable in present form. I just suggest to the authors to check the style of their papers before their next submissions in any journal.
Author Response
Thanks for your help, we found your comments and suggestions very helpful! We checked the style and corrected the last few issues.
with regards,
Jan Kavan
Reviewer 4 Report
I have no further comment.
Author Response
Thanks for your help with improving the manuscript. We appreciate your comments and suggestions!
with regards,
Jan Kavan